# ALIGNING VIDEOLLMS WITH VIDEO DIFFUSION FOR FINE-GRAINED TEMPORAL UNDERSTANDING

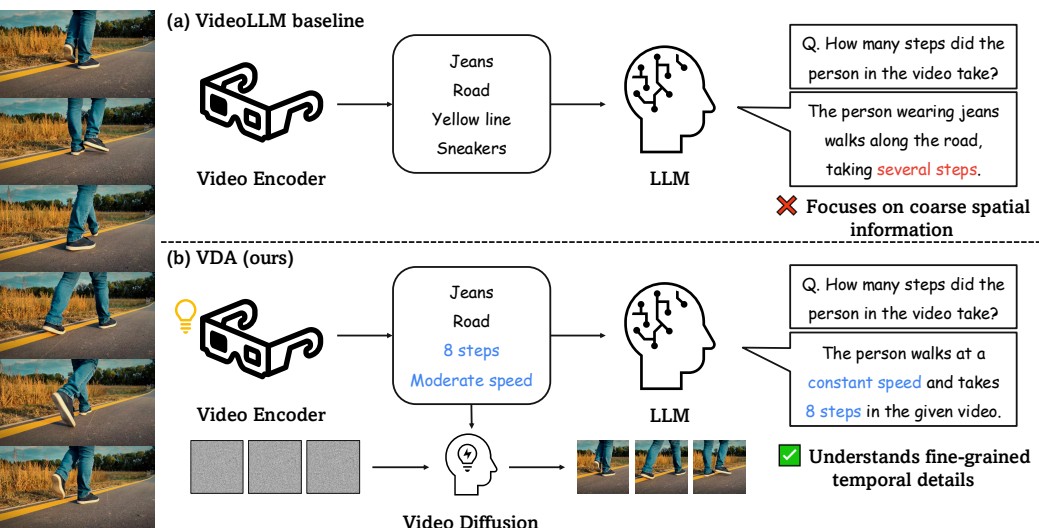

Figure 1: **Conceptual Illustration of VDA**. VideoLLMs perceive video content through video encoders, which often prioritize spatial or high-level semantic information, causing their output features to lack fine-grained temporal details. VDA addresses this limitation by leveraging a pretrained video diffusion model as an expert motion guide for the VideoLLM's video encoder, thereby enhancing its understanding of fine-grained motion.

## ABSTRACT

Video Question Answering (VideoQA) has traditionally revolved around tasks addressable by recognizing objects or simple events. However, the frontier of the field is increasingly pushing towards challenges that require reasoning about fine-grained motion and subtle temporal dynamics. This shift exposes a critical limitation in contemporary VideoLLMs, which often struggle to perceive these intricate dynamics. To address this, we introduce Video Diffusion Alignment (VDA), a framework that leverages the inherent ability of pretrained video diffusion models to represent intricate motion dynamics, thus enhancing motion representation learning. Our method steers a VideoLLM to focus on complex motion patterns by employing motion-centric knowledge from the diffusion model, resulting in more robust and detailed temporal features. Through extensive experiments, we show that VDA maintains competitive performance on traditional VideoQA benchmarks such as MSVD-QA and MSRVTT-QA, while boosting scores on MotionBench, a benchmark specifically designed for fine-grained motion understanding. This result is observed across three different VideoLLMs with different architectures, confirming the generality of our approach.

# 1 INTRODUCTION

The successful integration of Large Language Models (LLMs) with vision encoders has led to rapid advancements in multimodal models, particularly for static images (Liu et al., 2023; Li et al., 2023; Chen et al., 2024b; Team et al., 2025; Yang et al., 2025). A principal focus of recent research has been to extend this paradigm to the video domain (Lin et al., 2024; Li et al., 2024; Cheng et al., 2024; Wang et al., 2024c; Dai et al., 2023; Maaz et al., 2024b), which introduces the unique challenge of understanding not just static objects, but the temporal dynamics connecting them. To address this, two primary approaches have emerged: one leverages the capabilities of established image encoders by applying them to sparsely sampled frames (Li et al., 2024; Cheng et al., 2024; Wang et al., 2024c; Dai et al., 2023), while the other develops specialized, video-native architectures to capture temporal information more directly (Jin et al., 2024; Lin et al., 2024; Maaz et al., 2024b). Advancements in both streams have powered models capable of complex reasoning tasks that require integrating semantic information across multiple frames.

Despite this progress, the frontier of video understanding is pushing beyond recognizing general events towards a more distinct challenge: the perception and reasoning of fine-grained, motion-centric details. This shift is driven by a growing demand for models to move beyond the overall gist of a video and precisely track content over time. Such motion-centric tasks remain challenging for frontier models, a fact highlighted by recent benchmarks such as MINERVA (Nagrani et al., 2025), MotionBench (Hong et al., 2025), and V-STaR (Cheng et al., 2025). For instance, the MINERVA benchmark (Nagrani et al., 2025) identified repetition counting as a core weakness for Video Large Language Models (VideoLLMs). This limitation makes it difficult to answer questions like, "How many times did the player bounce the basketball before taking the shot?" which demands continuous and detailed observation. Therefore, this focus on temporal precision, capturing both intermittent events and repetitive actions, represents the next critical step in video comprehension, moving from high-level recognition to robust and detailed temporal awareness.

This emerging challenge exposes a key limitation in contemporary VideoLLMs, as their learned representations are not inherently attentive to fine-grained temporal details. The root of this problem often lies in the pretraining objectives used for their video encoders. These encoders are typically trained to produce a single, holistic representation for a video clip that aligns with a corresponding text caption or an action class (Xu et al., 2021; 2024; Wang et al., 2024c; Zhu et al., 2024). While effective for capturing the overall semantic gist of an event, this paradigm incentivizes the model to abstract away from, rather than preserve, temporal minutiae. Events like a briefly flashing light, or the precise number of repetitions in an action, are typically treated as noisy details irrelevant to the primary objective of high-level semantic alignment. This bias toward high-level semantics is often deepened by the nature of the datasets used for pretraining. Most large-scale video captioning datasets (Carreira & Zisserman, 2017; Zhou et al., 2018; Bain et al., 2021; Hendricks et al., 2018; Wang et al., 2019; Xu et al., 2017; Krishna et al., 2017) are themselves designed to support this holistic alignment, and thus primarily describe objects and general events, offering sparse supervisory signals for the subtle dynamics of movement. Consequently, while VideoLLMs are proficient at identifying entities across time, their representations frequently lack the precision to track momentary events or count occurrences.

On the other hand, recent advancements have highlighted the potential of using models trained on high-fidelity reconstruction to enhance visual representations. A prime example is the recent work by DIVA (Wang et al., 2025), which demonstrated that fine-tuning an image encoder with a diffusion model's objective leads to representations that are more attentive to fine-grained visual details. While these findings address static visual details, our objective is to enhance the understanding of motion and complex temporal dynamics, for which a supervisory signal from the static domain is insufficient. To bridge this gap, we propose that the necessary supervisory signal can be sourced by extending this reconstruction principle to video, finding it embodied in pretrained video diffusion models (Guo et al., 2024; Ho et al., 2022; Google DeepMind, 2025). Video diffusion models utilize a reconstruction-based objective, in contrast to the compression-focused approach of typical VideoLLMs. To successfully regenerate a video, the model must learn to encode the very details that semantic compression discards, including transient events and action frequencies. Consequently, their learned representations serve as a rich source of the precise temporal information that VideoLLMs overlook.

Drawing on this insight, we introduce Video Diffusion Alignment (VDA), a post-training alignment framework designed to infuse the VideoLLM's video encoder with a sensitivity to the precise temporal information. Our approach leverages a pretrained video diffusion model to enhance the video encoder, compelling it to learn motion-rich features through a diffusion loss. Our main contributions can be summarized as follows.

- We propose VDA, a post-training framework that enhances the motion perception of a VideoLLM's video encoder by aligning its representation with a video diffusion model.

- We introduce a method to provide a targeted learning signal by disentangling video captions into static (scene-based) and dynamic components, which compels the model to focus specifically on motion information.

- Our framework is designed to be flexible, improving the motion perception of VideoLLMs even in scenarios where text captions are unavailable.

- Through extensive experiments, we demonstrate that our method yields improved performance on a diagnostic benchmark for fine-grained motion understanding (i.e., Motion-Bench) while maintaining performance on general VideoQA datasets, consistently across three different model classes.

## 2 RELATED WORK

**Vision-Language models** Video Large Language Models (VideoLLMs) derive visual representations using two primary architectural approaches: processing sparsely sampled frames with static image encoders (Cheng et al., 2024; Wang et al., 2024c) or incorporating video-native modules (Lin et al., 2024; Yan et al., 2021; Jin et al., 2024). Despite these architectural advancements, a critical limitation persists, rooted in their training objectives. The pretraining for these vision backbones involves datasets with holistic annotations like Kinetics (Carreira & Zisserman, 2017), or relies on video-text contrastive learning on large-scale datasets (Xu et al., 2021; Sun et al., 2022; Zhao et al., 2023; Zhang et al., 2023). Both objectives incentivize the model to learn a single, holistic representation that captures the overall semantic gist of a video. This focus on high-level abstraction systematically filters out fine-grained temporal details, such as the precise count of an action or the presence of a fleeting event, as they are often irrelevant for minimizing the primary training loss (Nie et al., 2024; Wang et al., 2024a; Tu et al., 2025; Li et al., 2025).

**Fine-grained temporal reasoning in video tasks** The aforementioned limitations of VideoLLMs are becoming increasingly salient with the recent emergence of more demanding video understanding benchmarks. Traditional Video Question Answering (VideoQA) datasets, such as MSVD-QA and MSRVTT-QA (Xu et al., 2017), were pivotal in advancing the field but often contain questions that can be answered by identifying key objects or general scene context, sometimes without a detailed temporal understanding. Recognizing this gap, the research community has introduced a new wave of benchmarks designed to probe deeper temporal capabilities. This trend has intensified with recent benchmarks specifically targeting more complex and fine-grained analysis. For instance, benchmarks such as V-STaR (Cheng et al., 2025) and MINERVA (Nagrani et al., 2025) move beyond simple event ordering to evaluate complex reasoning, assessing not just final answers but the intermediate spatio-temporal and logical steps required. MotionVid-QA dataset (Du et al., 2025) offers high-fidelity annotations with high human preference to evaluate motion understanding in videos. Others focus on the nuances of motion itself; ActionArt (Peng et al., 2025) provides highly detailed annotations of human limb movement, while MotionBench (Hong et al., 2025) focuses specifically on a model's ability to perceive subtle, kinematic details. The proliferation of these benchmarks underscores a clear need in the field: models must evolve beyond high-level semantic summarization to develop a robust and precise awareness of fine-grained temporal events.

**Diffusion models** Diffusion models are a class of generative models operating on a denoising principle (Sohl-Dickstein et al., 2015; Ho et al., 2020; Song et al., 2021). The core mechanism involves training a model to iteratively reverse a noise-corruption process, allowing it to reconstruct clean data such as an image or a video from a noisy input. This high-fidelity reconstruction objective requires the model to learn a detailed internal representation of the data distribution.

**Representation alignment with generative models** Aligning internal representations between separately trained neural networks has recently emerged as a powerful technique for knowledge transfer and feature enhancement. This principle is demonstrated across various domains by methods such as Representation Probing and Alignment (REPA) (Yu et al., 2025), which transfers capabilities from self-supervised encoders to diffusion models, and knowledge insulation techniques that utilize LLM's semantic features to train continuous actions (Driess et al., 2025). The principle also encompasses using diffusion models as a source of rich supervisory features. In the image domain, DIVA (Wang et al., 2025) successfully utilized a pretrained diffusion model to refine the features of a vision encoder of an LLM. This principle was extended to the temporal domain by DIVOT (Ge et al., 2025), which successfully used a diffusion-based objective to train a new, specialized video encoder from scratch, demonstrating a strong capability for capturing detailed motion. We build on the same insight of using a diffusion model as a rich supervisory signal for motion. However, our work addresses a different yet complementary challenge: how to adapt this principle to efficiently enhance a large-scale, pretrained VideoLLM, rather than training a new component. We focus on a fine-tuning framework that aligns the vision encoder with a diffusion model, seeking to instill fine-grained motion details into the model's existing encoder without requiring full retraining or compromising its extensive, pre-existing general-purpose knowledge.

## 3 METHOD

Our proposed method, VDA, enhances the motion understanding of a VideoLLM by aligning motion-centric representations from a video diffusion with its vision encoder $E_\phi$. We accomplish this via a three-stage training procedure that leverages a pretrained video diffusion model as an expert motion assistant. This section first outlines the prerequisite architectures and our overall approach, then details each of the three training stages.

### 3.1 PRELIMINARIES

Our framework is designed to be broadly applicable to various VideoLLM architectures that possess a distinct module for temporal understanding. This includes models that explicitly separate image and motion encoders, such as VideoLaVIT (Jin et al., 2024), as well as those employing a unified video encoder, such as VideoLLaVA (Lin et al., 2024). Since VDA can enhance the *motion* understanding of both model classes (e.g. unified video encoder of VideoLLaVA (Lin et al., 2024), motion encoder of VideoLaVIT (Jin et al., 2024)), we use the term "video encoder" to acknowledge the two different types of video encoders henceforth. A typical VideoLLM processes an input video $V \in \mathbb{R}^{B \times T \times H \times W \times C}$ to generate a text-based answer $A$ for a given question $Q$. The video encoder $E_\phi$ first extracts a sequence of visual features $F_V = E_\phi(V) \in \mathbb{R}^{T \times d_v}$. These features are fed into an LLM along with the tokenized question to produce the final answer. We formulate this process as:

$$F_V = E_\phi(V), \quad A = \text{VideoLLM}(F_V, Q) \tag{1}$$

The second key component of our method is a pretrained, text-conditioned video diffusion model (Guo et al., 2024; Wang et al., 2024b; Chen et al., 2024a). Without loss of generality, we assume that the diffusion model operates in the latent space, which encodes pixels $x_0 \in \mathbb{R}^{B \times T \times H \times W \times C}$ to low-dimensional latents (Rombach et al., 2022) $z_0 \in \mathbb{R}^{B \times T \times h \times w \times c}$ with an encoder $\mathcal{E}$. The model learns to predict the noise $\epsilon$ added to a clean video latent $z_0$ at a specific timestep $t$, conditioned on a text description $c_{\text{text}}$. The diffusion model, parameterized by $\theta$, minimizes the following denoising objective:

$$\mathcal{L}_{\text{DM}} = \mathbb{E}_{z_t \sim q(z_t|z_0), z_0 \sim \mathcal{E}(x_0), x_0 \sim p_{\text{data}}, c_{\text{text}}, \epsilon \sim \mathcal{N}(0,I), t \sim \mathcal{U}[0,1]} \left[ ||\epsilon - \epsilon_\theta(z_t, t, c_{\text{text}})||^2 \right] \tag{2}$$

where $z_t$ is the noisy video sampled at time $t$ with the forward diffusion kernel $q(z_t|z_0)$ (Ho et al., 2020).

### 3.2 VDA: OVERALL ARCHITECTURE

The central goal of VDA is to enrich the video encoder $E_\phi$ so that its output features $F_V$ are more sensitive to fine-grained motion. To achieve this, we create a connection between the VideoLLM's

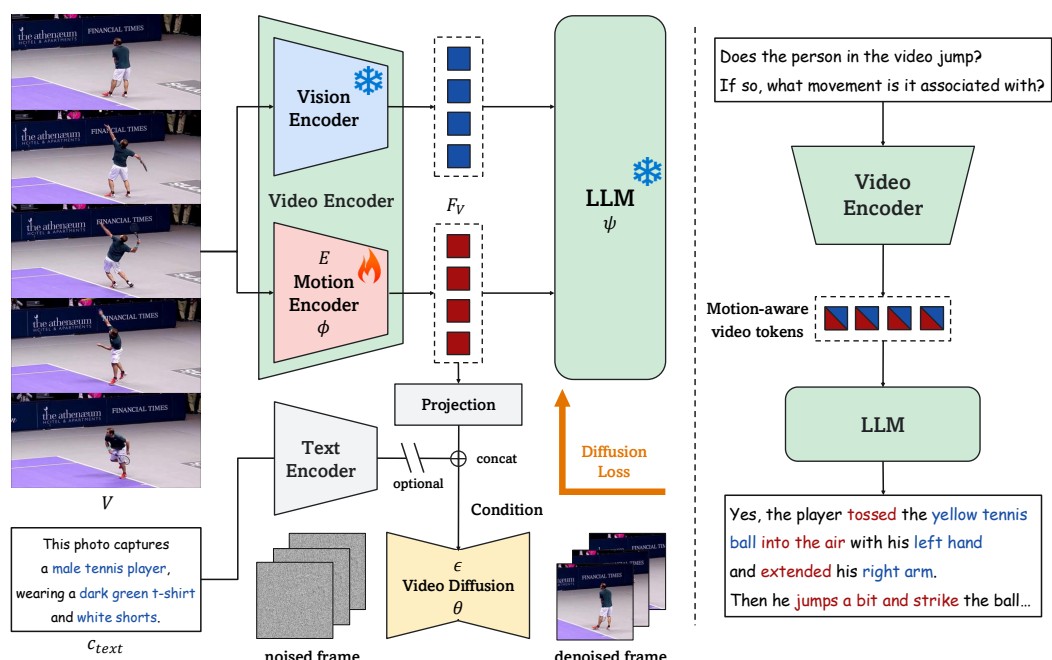

Figure 2: **Overall Architecture of VDA**. The figure shows an illustration of the application of VDA in VideoLaVIT. Our approach utilizes both spatial information from captions and features from a video encoder as conditioning inputs for a pretrained video diffusion model. The resulting diffusion loss compels the video encoder to focus on the fine-grained temporal details that captions alone cannot provide, ensuring successful video reconstruction. This aligned video encoder subsequently improves the VideoLLM's overall understanding of fine-grained motion.

video encoder and the pretrained video diffusion model. Specifically, the features $F_V$ extracted from $E_\phi$ are passed through a linear projection layer and used as an additional condition for the diffusion model, such that the gradients from the diffusion loss can backpropagate to enhance the representation of the video encoders (See Fig. 2 for an illustration). This allows the diffusion model's powerful understanding of motion dynamics to guide the training of $E_\phi$.

The entire training process is divided into three distinct stages, each with a specific goal:

- Stage 1 (Diffusion Model Alignment): The video diffusion model and the projection layer are trained to comprehend the feature space of the VideoLLM's video encoder.

- Stage 2 (Video Encoder Representation Alignment): The video diffusion model provides a supervisory signal to align the representation with the video encoder.

- Stage 3 (LLM Adaptation): The LLM is fine-tuned to properly interpret the new, enhanced visual features from the aligned encoder.

### 3.3 STAGE 1: PREPARING THE DIFFUSION AS A MOTION GUIDE

The goal of this initial stage is to align the two disparate components: the VideoLLM's video encoder and the pretrained diffusion model. This alignment ensures that the diffusion model can interpret the visual features $F_V$ as a meaningful condition, enabling an effective use in later stages as a meaningful *assistant* model for the representation alignment. See Fig. 3a for an illustration. During this stage, the video encoder $E_\phi$ is kept frozen. We introduce a new, trainable linear projection layer and apply Low-Rank Adaptation (LoRA) (Hu et al., 2022) to the U-Net of the diffusion model. The diffusion model is then conditioned on both the projected visual features and a scene-based text description $\mathbf{c}_{\text{text}}$, which is generated by an LLM to contain only the spatial information (see Appendix A for details). The model is trained by optimizing the denoising loss, updating only the

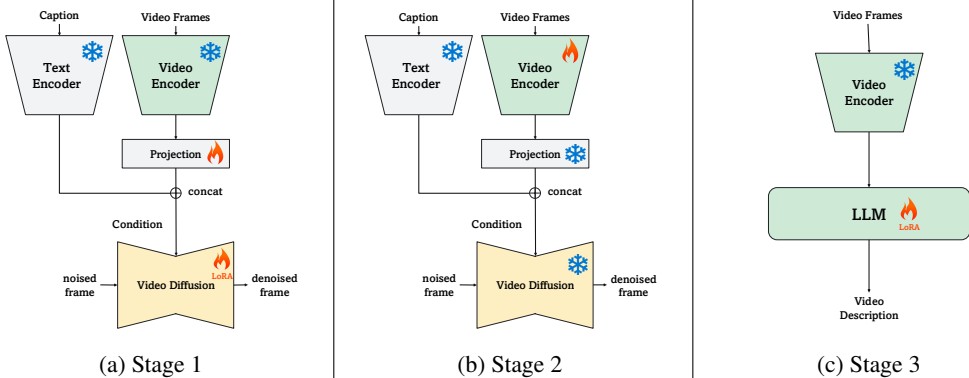

(a) Stage 1        (b) Stage 2        (c) Stage 3

Figure 3: **A detailed illustration of our three-stage training procedure.** (a) Stage 1: Diffusion Model Alignment. With the video encoders frozen, a projection layer and the LoRA adapters of the video diffusion model are trained to comprehend the feature space of the VideoLLM's video encoder. (b) Stage 2: Video Encoder Representation Alignment. The diffusion model and projection layer are subsequently frozen, and the denoising loss is used as a supervisory signal to exclusively train the video encoder. (c) Stage 3: LLM Adaptation. Finally, the refined video encoder is frozen, and the LLM is fine-tuned using LoRA to adapt to the enhanced motion-aware features.

weights of the linear layer and the LoRA adapters. The optimization objective reads:

$$\min_{W_{\text{proj}}, \Delta\theta_{\text{LoRA}}} \mathbb{E}_{\boldsymbol{V}, \mathbf{z}_t, \mathbf{z}_0, \mathbf{x}_0, \mathbf{c}_{\text{text}}, \boldsymbol{\epsilon}, t} \left[ ||\boldsymbol{\epsilon} - \boldsymbol{\epsilon}_{\theta + \Delta\theta_{\text{LoRA}}}(\mathbf{z}_t, t, \mathbf{W}_{\text{proj}}(E_{\boldsymbol{\phi}}(\boldsymbol{V})), \mathbf{c}_{\text{text}})||^2 \right] \quad (3)$$

### 3.4 STAGE 2: REPRESENTATION ALIGNMENT OF VIDEO ENCODER

With the feature spaces aligned, the second stage uses the aligned diffusion model as an assistant to improve the video encoder. In this phase, the diffusion model, the LoRA adapters, and the linear projection layer are all frozen. The only trainable component is the video encoder $E_{\boldsymbol{\phi}}$ itself. The training process remains the same: the denoising loss from the diffusion model as in equation 3. However, the gradients from this loss are now backpropagated all the way to the video encoder $E_{\boldsymbol{\phi}}$. See Fig. 3b for an illustration. This process compels $E_{\boldsymbol{\phi}}$ to produce features that are more informative for the diffusion model's denoising task—and thus, richer in fine-grained motion details. The optimization process can be expressed as below

$$\boldsymbol{\phi}^* = \arg\min_{\boldsymbol{\phi}} \mathbb{E}_{\boldsymbol{V}, \mathbf{z}_t, \mathbf{z}_0, \mathbf{x}_0, \mathbf{c}_{\text{text}}, \boldsymbol{\epsilon}, t} \left[ ||\boldsymbol{\epsilon} - \boldsymbol{\epsilon}_{\theta + \Delta\theta_{\text{LoRA}}}(\mathbf{z}_t, t, \mathbf{W}_{\text{proj}}(E_{\boldsymbol{\phi}}(\boldsymbol{V})), \mathbf{c}_{\text{text}})||^2 \right]. \quad (4)$$

### 3.5 STAGE 3: ADAPTING THE LLM TO THE ENHANCED ENCODER

The final stage ensures that the VideoLLM can effectively utilize the newly enhanced video encoder. In this phase, the aligned video encoder $E_{\boldsymbol{\phi}^*}$ from Stage 2 is frozen. We then perform end-to-end fine-tuning on the video captioning datasets, targeting to generate an appropriate explanation $A_{\text{gt}}$ for a given question $Q$. See Fig. 3c for the illustration. To maintain efficiency, we apply LoRA to the LLM, updating only its adapter weights. This step adapts the LLM parameters $\psi$ of the VideoLLM to the improved visual features, enabling it to better leverage the enhanced motion understanding for the final reasoning task. The training objective is the standard cross-entropy loss for the task:

$$\min_{\Delta\psi_{\text{LoRA}}} \mathbb{E}_{\boldsymbol{V}, Q, A_{\text{gt}}} \left[ -\log P_{\psi + \Delta\psi_{\text{LoRA}}}(A_{\text{gt}} | E_{\boldsymbol{\phi}^*}(\boldsymbol{V}), Q) \right] \quad (5)$$

## 4 EXPERIMENTS

### 4.1 EXPERIMENTAL SETUP

**Datasets** For computational efficiency, we utilize 10,000 videos, randomly sampled from OpenVid-1M (Nan et al., 2025) in a manner that preserves its statistical properties, to train our

Table 1: **Result of VDA on Video-QA benchmarks.** Comparison of our method (VDA) against each baseline group, showing performance gains. [†] denotes our implementation of the baseline models. Positive changes are in red, and negative changes are in blue.

| Model | MotionBench | MSVD | | MSRVTT | |
|---|---|---|---|---|---|
| | | Acc. | Score | Acc. | Score |
| VideoLaVIT[†] | 38.53 | 65.07 | 3.25 | 52.30 | 3.21 |
| **+VDA** | **39.80** (+1.27) | **65.46** (+0.39) | **3.22** (-0.03) | **52.90** (+0.60) | **3.31** (+0.10) |
| Video-LLaVA[†] | 30.51 | 63.60 | 3.16 | 55.84 | 3.31 |
| **+VDA** | **32.49** (+1.98) | **63.79** (+0.19) | **3.12** (-0.04) | **56.21** (+0.37) | **3.33** (+0.02) |
| VideoGPT+[†] | 46.51 | 72.54 | 3.61 | 60.65 | 3.59 |
| **+VDA** | **47.86** (+1.35) | **73.13** (+0.59) | **3.60** (-0.01) | **61.18** (+0.53) | **3.60** (+0.01) |

model in all three stages. We evaluate the zero-shot performance of our method on three VideoQA benchmarks: MSVD-QA, MSRVTT-QA (Xu et al., 2017), and MotionBench (Hong et al., 2025). MSVD-QA and MSRVTT-QA are standard benchmarks covering a broad range of questions, while MotionBench is a diagnostic benchmark specifically designed to assess the understanding of fine-grained motion. For the MSVD-QA and MSRVTT-QA benchmarks, the original papers reported results using a GPT-3.5-based evaluator. As this is no longer available, we re-evaluated all re-produced baselines using the more capable GPT-4o-based evaluator, while keeping the evaluation prompt identical (Jin et al., 2024; Maaz et al., 2024a). For the evaluation on MotionBench, we utilize the DEV-set.

**Baseline models** The general applicability of VDA is demonstrated on three distinct and popular VideoLLMs: VideoLaVIT (Jin et al., 2024), VideoLLaVA (Lin et al., 2024), and VideoGPT+ (Maaz et al., 2024b). Our method is tailored to the unique visual encoding strategy of each model. For VideoLaVIT, which features separate image and motion encoders, only the motion encoder is trained using our method. For models with a single video encoder, our method operates on their respective visual backbones: the LanguageBind (Zhu et al., 2024) for VideoLLaVA and the InternVideo2 (Wang et al., 2024c) for VideoGPT+. While the number of input frames for each baseline model follows their original settings, the number of input frames for the diffusion model within our method consistently uses 24 frames.

**Implementation details** In VDA, we use the pretrained video diffusion model, AnimateDiff (Guo et al., 2024). We conduct our three-stage training with a batch size of 1 and a learning rate of 1e-5. For the LoRA-based training in Stage 1 and Stage 3, we use a rank of 16. We conduct all experiments on 2 A6000 GPUs. For evaluation, we report the standard accuracy metric for all benchmarks.

## 4.2 MAIN RESULTS

We apply VDA to the three baseline models and evaluate them on the aforementioned VideoQA benchmarks. The results are summarized in Table 1. On MotionBench, the benchmark that directly evaluates fine-grained temporal understanding, all models trained with VDA show substantial performance improvements. This consistent gain across diverse model architectures validates our core hypothesis: by leveraging the rich, dynamic priors from a video diffusion model, we can effectively enhance VideoLLM encoders to perceive subtle motion cues that they would otherwise miss. This highlights the precision of our method in enhancing the targeted motion understanding capability.

This targeted refinement of motion-specific capabilities does not come at the cost of degrading general reasoning abilities. On the general VideoQA benchmarks, MSVD-QA and MSRVTT-QA, models post-trained with VDA achieve comparable performance to their original counterparts. The representation alignment successfully enriches the visual encoder without causing catastrophic forgetting of other important visual concepts. The qualitative results illustrated in Table 2 again confirm the enhancement.

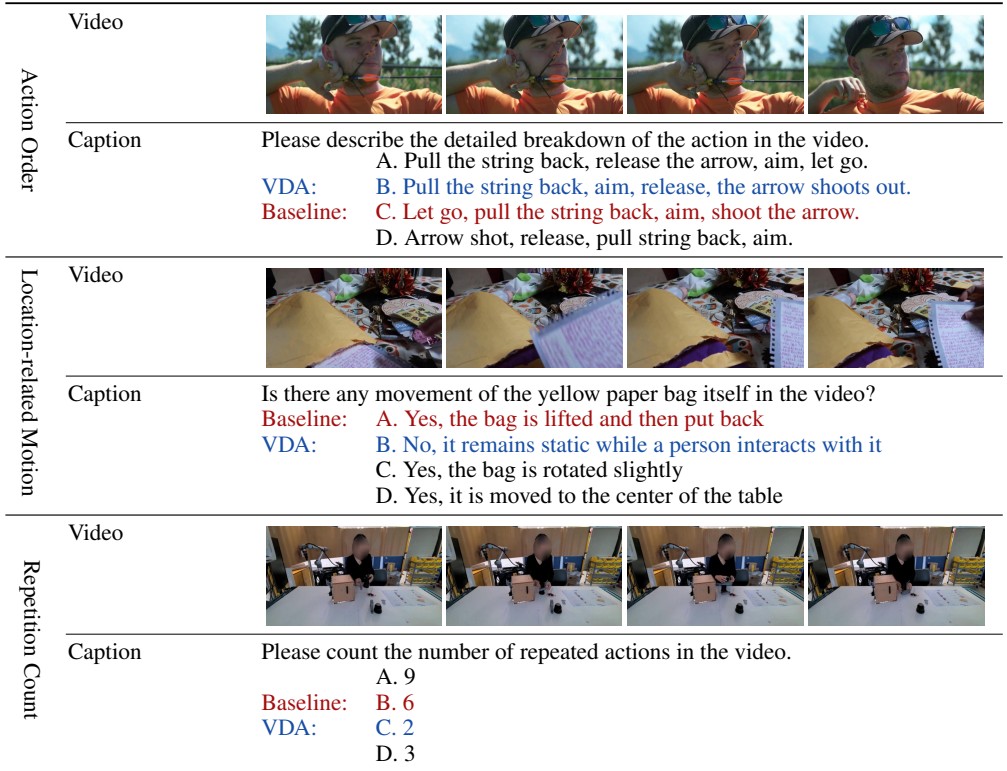

Table 2: **Qualitative result.** Comparison of our model and the VideoLaVIT baseline on three categories from MotionBench. The results highlight our model's capability in temporal reasoning. It correctly interprets explicit sequences in Action Order, Repetition Count and Location-related Motion.

## 4.3 ABLATION STUDIES

**Ablating learned temporal knowledge of the video diffusion model**  To precisely isolate the contribution of temporal knowledge, we conduct a controlled experiment where we replace our video diffusion model with its image backbone, Stable Diffusion v1.5 (Rombach et al., 2022). This configuration effectively removes the temporal modeling components while retaining the same spatial priors, as AnimateDiff (Guo et al., 2024) is derived from Stable Diffusion v1.5. As shown in Table 3, we observe that performing alignment with an image diffusion model *degrades* the performance on MotionBench, while it enhances the performance slightly on other benchmarks. In contrast, VDA achieves much improved performance in MotionBench. This result leads to two key conclusions. First, it confirms that the temporal knowledge inherent in the video diffusion is crucial for our training pipeline, as its spatially-focused backbone alone is insufficient. Second, it suggests that the performance improvement does not simply stem from refining fine-grained spatial details, but originates directly from an enhanced understanding of motion.

**Impact of training dataset scale**  We evaluated the impact of training data scale on VDA using the subset of OpenVid dataset, ranging from 1k to 100k samples (Figure 4). On our primary benchmark, MotionBench, scaling the dataset from 1k to 10k samples resulted in a substantial performance improvement, while the subsequent increase to 100k samples yielded a modest gain, with a drop in performance in MSRVTT. Given the considerable computational resources required to expand the dataset to 100k samples, we adopted the 10k sample configuration for all other experiments to ensure computational efficiency.

**Text conditioning enhances LLM focus on motion**  To isolate the contribution of text conditioning, we utilize the diffusion model using only visual features from the VideoLLM's encoder, with the corresponding text caption removed. As presented in Table 3, this alignment with visual features alone yields a modest gain over the baseline. Furthermore, the inclusion of text captions, which

Table 3: **Comprehensive ablation studies.** We analyze the impact of different components on model performance. The highlighted row represents our main model configuration. Results are top-1 accuracy (%).

| Model | Configuration | | Metrics (Top-1 Acc. %) | | |
|---|---|---|---|---|---|
| | **Text Conditioning** | **Diffusion Backbone** | **MSVD** | **MSRVTT** | **MotionBench** |
| VideoLaVIT[†] | - | - | 65.07 | 52.30 | 38.53 |
| +VDA | × | AnimateDiff | 65.44 | 52.49 | 39.27 |
| +VDA | ✓ | StableDiffusion 1.5 | 65.33 | 52.45 | 37.93 |
| +VDA | ✓ | AnimateDiff | **65.46** | **52.90** | **39.80** |

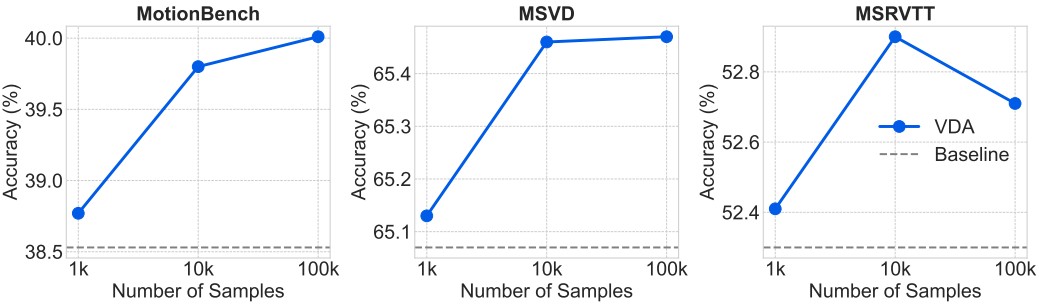

Figure 4: **Ablation on the post-training data scale.** Performance of VDA (Video-LaVIT) when trained on 1k, 10k, and 100k samples from OpenVid. We report top-1 accuracy (%) on VQA benchmarks.

constitutes our full method, results in a larger performance gain. Hence, VDA serves as a flexible framework that can be used to enhance the representation of VideoLLMs regardless of whether or not the video captions are available. When the captions are available, we can further leverage this information to help the model focus on motion cues.

## 5 CONCLUSION

In this work, we introduce VDA, a novel post-training framework designed to improve fine-grained motion understanding in VideoLLMs. This work identifies the core challenge as a tendency of video encoders to discard the temporal minutiae essential for fine-grained motion reasoning. VDA addresses this by leveraging a pretrained video diffusion model as an expert motion assistant. Through a three-stage process, our method refines the video encoder's representation space, compelling it to capture the rich, dynamic details inherent in the diffusion model's generative knowledge. The approach is predicated on the principle that the reconstruction-based objective of diffusion models serves as a rich source of supervisory signals for the precise temporal information that VideoLLMs typically overlook. Extensive experiments on diverse VideoLLM architectures validate this approach. VDA demonstrates performance gains on MotionBench, a benchmark designed specifically for fine-grained motion, while maintaining competitive performance on general VideoQA benchmarks. These results establish that aligning VideoLLMs with video diffusion priors is a potent strategy for overcoming key perceptual bottlenecks, enabling a deeper understanding of how events unfold over time beyond simple gist recognition.

**Discussion** While VDA effectively enhances fine-grained motion understanding, the current implementation presents several limitations, which in turn suggest promising directions for future work. A primary limitation is the dependence on the pretrained video diffusion model, as any biases or weaknesses in this "motion guide" may be transferred to the VideoLLM's encoder. Second, the current three-stage training process is computationally intensive and complex to manage. These challenges motivate several future research directions. The first direction is streamlining the training process into a unified, one-stage pipeline. A second avenue explores broadening the method's applicability to VideoLLMs that leverage image encoders. These directions aim to improve the framework's operational efficiency and broaden its applicability, contributing to a practical and generalizable approach for enhancing motion understanding.

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

## A   EXPERIMENT DETAILS

**Used prompts**   The scene-based text descriptions ($c_{text}$) used for conditioning the diffusion model in Stage 1 are derived from the ground-truth (GT) captions of the pretraining dataset. To isolate spatial information, we utilize GPT-4 to parse the original GT captions. Specifically, GPT-4 is prompted to extract and rewrite a new caption containing only descriptions of static scenes, objects, and their spatial relationships, while explicitly removing any information related to temporal events or motion. Table 4 shows the prompt.

For the MSVD and MSRVTT benchmarks, we adopt the question and prompting strategies directly from the original VideoLaVIT and Video-ChatGPT papers. The complete text of the prompt can be found in Table 5.

---

**System prompt:**
You are an intelligent chatbot designed to rephrase and summarize input text. Your task is to condense a given paragraph describing a video into one or two sentences, preserving its spatial key information.
**User prompt:**
Please summarize the following video explanation into one or two sentences:
<GT_CAPTION>

---

Table 4: Prompts used for the caption extraction.

---

**System prompt:**
You are an intelligent chatbot designed for evaluating the correctness of generative outputs for question-answer pairs.
Your task is to compare the predicted answer with the correct answer and determine if they match meaningfully. Here's how you can accomplish the task:
——
##INSTRUCTIONS:
- Focus on the meaningful match between the predicted answer and the correct answer.
- Consider synonyms or paraphrases as valid matches.
- Evaluate the correctness of the prediction compared to the answer.
**User prompt:**
Please evaluate the following video-based question-answer pair:
Question: <Question>
Correct Answer: <Ground Truth>
Predicted Answer: <Prediction>
Provide your evaluation only as an integer value between 1 and 5, with 5 indicating the highest quality.
Please generate the response in the form of a Python dictionary string with keys 'pred' and 'score', where value of 'pred' is a string of 'yes' or 'no' and value of 'score' is in FLOAT, not STRING.
DO NOT PROVIDE ANY OTHER OUTPUT TEXT OR EXPLANATION. Only provide the Python dictionary string.
For example, your response should look like this: {'pred': 'yes', 'score': 4.8}.

---

Table 5: Prompts used for MSVD, MSRVTT evaluation (Jin et al., 2024; Maaz et al., 2024a).

## B   ABLATION ON PROJECTION LAYER

We analyzed the performance of the projection layer based on its architecture. We compare two configurations: one using an MLP block (composed of two linear layers with an intermediate GELU

activation, similar to the ViT structure) and the other using a single linear layer. Table 6 shows that the performance difference between the two approaches was not substantial. However, the single linear layer configuration yields slightly better results. This suggests that a simpler, direct linear transformation is more effective at aligning the feature representations for the downstream task than a more complex non-linear projection.

Table 6: **Impact of projection layer** We compare the performance of VDA trained with a single linear layer and a MLP block containing two linear layers and GELU. Results are top-1 accuracy (%) on zero-shot VQA benchmarks.

| Method | Projection | MSVD-QA | MSRVTT-QA | MotionBench |
|---|---|---|---|---|
| VideoLaVIT | - | 65.07 | 52.30 | 38.53 |
| VDA | MLP | 65.13 | 52.40 | 39.47 |
| | Linear | 65.46 | 52.90 | 39.80 |

## C    ABLATION ON TRAINING STAGES

To analyze the contribution of each phase in our three-stage training process, we conduct an ablation study focusing on the final LLM adaptation stage. As shown in Table 7, we evaluate the model's performance after completing only the first two stages (Stage 1+2) and compare it against the full pipeline which includes Stage 3. The results reveal a performance degradation after applying only Stage 1 and 2. This indicates the LLM, which has not been trained on new representations, fails to properly interpret them for downstream reasoning tasks. However, upon completing Stage 3, where the LLM is fine-tuned using LoRA to adapt to the enhanced encoder, performance not only recovers but surpasses the original baseline across all benchmarks. This confirms that Stage 3 is an essential step that bridges the gap between the refined encoder and the language model. It enables the VideoLLM to effectively leverage the new motion-aware features, realizing the full potential of our framework.

Table 7: **Necessity of the LLM Adaptation Stage (Stage 3).** We compare the performance after Stage 2 against the full three-stage process. Results are top-1 accuracy (%) on zero-shot VQA benchmarks.

| Method | Stage | MSVD-QA | MSRVTT-QA | MotionBench |
|---|---|---|---|---|
| VideoLaVIT | - | 65.07 | 52.30 | 38.53 |
| VDA | 1+2 | 63.49 | 49.98 | 38.27 |
| | 1+2+3 | 65.46 | 52.90 | 39.80 |

## D    LLM USAGE

In this work, we utilized Large Language Models (LLMs) for several aspects of the research and writing process. Specifically, we used Gemini 2.5 Pro for the following purposes:

- Writing Assistance: We used the LLM to proofread and refine the language in the introduction and conclusion sections for clarity and grammatical correctness.

All final claims, methodologies, and conclusions presented in this paper were critically reviewed and validated by the human authors.

