# OpenReview forum: "Aligning VideoLLMs with Video Diffusion for Fine-Grained Temporal Understanding"
_ICLR.cc/2026/Conference — ICLR 2026 Conference Withdrawn Submission_

### Official Review · Reviewer_pWkh · 2025-10-15

**Soundness:** 2
**Presentation:** 3
**Contribution:** 2
**Rating:** 4
**Confidence:** 4

**Summary:**

This paper introduces Video Diffusion Alignment (VDA), a post-training framework designed to enhance the fine-grained temporal understanding of Video Large Language Models (VideoLLMs), which often struggle to perceive subtle motion details. The core idea is to leverage a pretrained video diffusion model as a "motion expert" to guide the VideoLLM's video encoder. Through a three-stage training process, VDA aligns the video encoder's feature space with the diffusion model, using the diffusion's reconstruction loss as a supervisory signal to compel the encoder to produce representations rich in temporal minutiae. Finally, the LLM is fine-tuned to adapt to these enhanced features.

**Strengths:**

* The paper's focus on fine-grained motion, which is a key challenge in the video domain that deserves more attention.

* The core idea of using a generative video diffusion model, which excels at high-fidelity reconstruction, to supervise and refine the representations of a VideoLLM's encoder makes sense. The logic that a model trained to reconstruct a video must inherently understand its fine-grained temporal details is intuitive.

**Weaknesses:**

* The central idea that reconstruction-based objectives can enhance representation learning has been explored in prior work (e.g., [1, 2, 3]). Consequently, this paper's contribution could be seen as a straightforward extension of these existing reconstructive tuning paradigms from the image domain to video, which may limit its perceived novelty.

* While the authors emphasize that their method captures motion, the experiments lack qualitative visualizations to directly support this claim at the feature level. The paper would be significantly strengthened by including visual evidence (e.g., attention maps over time) to illustrate how the aligned encoder learns to focus on motion-rich patterns.

* The experimental validation on general-purpose benchmarks feels somewhat limited. While MotionBench directly supports the main claim, relying only on MSVD-QA and MSRVTT-QA to demonstrate generalizability is not fully sufficient. Evaluating the model on a wider array of benchmarks (e.g., videomme. mvbench and etc) would provide a more robust assessment of the VDA framework's overall impact.

[1] Ross3D: Reconstructive Visual Instruction Tuning with 3D-Awareness

[2] X-Former: Unifying Contrastive and Reconstruction Learning for MLLMs

[3] RECONSTRUCTIVE VISUAL INSTRUCTION TUNING

**Questions:**

Please see Weaknesses

---

### Official Review · Reviewer_Qoqc · 2025-10-17

**Soundness:** 2
**Presentation:** 3
**Contribution:** 2
**Rating:** 2
**Confidence:** 5

**Summary:**

The paper introduces VDA, a post-training framework designed to improve the ability of Video Large Language Models (VideoLLMs) to understand fine-grained motion and temporal dynamics in videos. The key innovation is to use a pretrained video diffusion model as a "motion expert" to guide the video encoder of a VideoLLM, making it more sensitive to subtle and complex motion patterns that typical training objectives often overlook. The training framework includes 3 stages: 1) the video diffusion model is adapted to interpret the VideoLLM’s encoder features, 2) the video encoder is trained (using the diffusion loss) to produce features rich in motion details, 3) the LLM is fine-tuned to utilize the improved, motion-aware video features.
The model is then evaluated on three different datasets and three different VLMs without SoTA comparison.

**Strengths:**

The paper has the following strengths:

1 - The idea is interesting and motion understanding is indeed a real weakness for current VLMs.

2 - Using diffusion model to enforce motion encoding is a clever decision (although is not completely new as the authors claim, diffusion alignment has already been explored in images, but it counts as a novelty since it is being applied to videos for a different purpose).

3 - Application of the proposed framework in different architectures, including VideoLaVIT, VideoLLaVA, and VideoGPT+.

**Weaknesses:**

While the method is interesting, I believe the study itself is weak for the standard of ICLR for the following purposes:

1 - The method is about improving motion understanding in VLMs, and is tested in 3 datasets: MotionBench, MSVD, and MSRVTT. From the three benchmarks, only MotionBench does actually requires motion understanding while MSVD and MSRVTT are more about semantic understanding with motion almost completely irrelevant. With that said, the claims are being tested only in one dataset and the comparison with SoTA methods is lacking (to avoid GPT-3.5 dependency, the authors could use alternative benchmarks). This lack of more diverse benchmarks and SOTA comparisons, makes the claims not fully sustained.

2 - The method does not scale well with the data size. In Fig. 4 we see that 10x data gives 0.5% improvement on MotionBench and causes performance loss on MSRVTT. The causes can be 2: 1) the extra data does not provide much information (although this is difficult to believe given that extra 90K videos are used), 2) the method has limitations which prevents scaling. This argument has not been properly treated in the paper. It is important to understand weather this is a method limitation or not.

3 - The training in 3 stages is a bit problematic. Especially in second stage where you train only the vision encoder. While the other elements remain fixed and the entire framework is aligned with text features, training only the visual encoder causes the model to lose control of that alignment. However, considering the evaluation setup, this is difficult to measure since the benchmarks are multiple-choice QA rather than open world question. I think this is an important aspect because VLMs are of no use if they lose the ability to properly understand and generate language so maybe some revisiting to the method is needed.

While there are other minor weaknesses, given the time constrain for the rebuttal period, I am giving a weak reject evaluation. The study is very shallow and lacks the depth required by the standards of this conference. However, If the authors address my concerns with enough evidence to sustain their claims, I am willing to increase my score even to a positive one.

**Questions:**

Check the weakness section.

---

### Official Review · Reviewer_uwWm · 2025-10-20

**Soundness:** 3
**Presentation:** 2
**Contribution:** 2
**Rating:** 2
**Confidence:** 5

**Summary:**

The paper proposes Video Diffusion Alignment (VDA), a post-training framework that enhances VideoLLMs’ sensitivity to fine-grained motion. By aligning the video encoder with pretrained video diffusion models and disentangling static and dynamic cues, VDA improves motion representation learning, boosting performance on MotionBench while maintaining general VideoQA accuracy.

**Strengths:**

The paper is original in leveraging pretrained video diffusion models to enhance temporal sensitivity in VideoLLMs. The methodology is well-motivated and technically sound, with a clear design for disentangling static and dynamic components. The approach is flexible, broadly applicable, and experiments demonstrate consistent gains, highlighting both practical significance and clarity of contribution.

**Weaknesses:**

The necessity of Stage 1 is unclear, overall novelty is limited, figures and experiments lack alignment with claims, and comparisons with newer benchmarks and methods are missing.

**Questions:**

1. The paper proposes a three-stage post-training to enhance the video encoder’s motion representation. However, intuitively Stage 1 may be unnecessary, and the authors should analyze the necessity of each stage.

2. The overall idea is relatively simple, making it difficult to justify the publication of a high-quality paper based on its novelty.

3. Figure 1 does not clearly match the methodology, and datasets like MSVD and MSRVTT are relatively old. Experiments on more recent benchmarks and comparisons with the latest methods are needed.

4. In Figure 4, the performance on MSRVTT decreases as the training data increases; the authors should explain this result.

---

### Official Review · Reviewer_bbng · 2025-11-01

**Soundness:** 2
**Presentation:** 2
**Contribution:** 1
**Rating:** 4
**Confidence:** 3

**Summary:**

The authors introduce a novel method to improve fine-grained motion understanding in VLMs by leveraging video diffusion models using a 3 stage alignment process.

**Strengths:**

1. The authors propose a novel method to align video diffusion models with VLMs to improve their fine grained motion understanding.
2. The proposed method can be applied to different classes of VLMs, ones using a separate motion encoder or ones using a unified vision encoder.

**Weaknesses:**

1. Unfortunately, it seems the improvement as a result of the proposed alignment is marginal (<2 %) in case of Motion-bench and vanishingly small (<1%) in case of MSVD and MSR-VTT.

2. The authors should experiment with different alignment methods as well, instead of adapting the VLMs vision/motion encoder, they should also try to directly use the video diffusion features as LLM input to see if that improves results.


Overall the main drawback is that the results are quite underwhelming relative to the complexity of alignment related interventions introduced.

**Questions:**

see weaknesses.

---

### Note · Authors · 2025-11-13

I have read and agree with the venue's withdrawal policy on behalf of myself and my co-authors.